# Factors associated with pre-loss grief and preparedness in relatives of people with cancer during the COVID-19 pandemic: A cross-sectional study

**Viktoria Schmidt** *, **Julia Kaiser, Julia Treml, Anette Kersting**

Department of Psychosomatic Medicine and Psychotherapy, University of Leipzig, Leipzig, Germany

* Viktoria.Schmidt@medizin.uni-leipzig.de

## Abstract

### Objectives

Before the loss of a loved one to cancer, relatives have time to adapt to the impending death. However, due to the current COVID-19 pandemic, adjustment to an imminent death may be more difficult. This study investigates factors related to pre-loss grief and preparedness during the COVID-19 pandemic and their relationship with COVID-19 related fears.

### Methods

Data of 299 participants from a cross-sectional study was used. Participants were included if they were relatives of people with cancer, spoke German and were at least 18 years. Multivariate linear regression analyses were conducted to measure the relationship between predictors (dysfunctional coping, emotion-focused coping, problem-focused coping, attachment anxiety, attachment avoidance, COVID-19 related fears, prognosis, perceived depth of the relationship, perceived conflict in the relationship, health status) and pre-loss grief, preparedness for caregiving and preparedness for death as the dependent variables.

### Results

Perceived depth (β = .365, $p$ < .001), COVID-19 related fears (β = .141, $p$ = .002), prognosis for death (β = .241, $p$ < .001), dysfunctional coping strategies (β = .281, $p$ < .001) and emotion-focused coping strategies (β = -.320, $p$ < .001) significantly predicted pre-loss grief. Prognosis for death (β = .347, $p$ < .001), dysfunctional coping strategies (β = -.229, $p$ < .001), emotion-focused coping strategies (β = .242, $p$ < .001), COVID-19 related fears (β = -.112, $p$ = .037) and health status (β = .123, $p$ = .025) significantly predicted preparedness for death. Dysfunctional coping (β = -.147, $p$ = .009), problem-focused coping (β = .162, $p$ = .009), emotion-focused coping (β = .148, $p$ = .017), COVID-19 related fears (β = -.151, $p$ = .006), attachment anxiety (β = -.169, $p$ = .003), perceived conflict in the relationship with the patient with cancer (β = -.164, $p$ = .004), perceived depth in the relationship (β = .116, $p$ = .048) and health status (β = .157, $p$ = .003) significantly predicted preparedness for caregiving.

**Data Availability Statement:** By instruction of the Office for Data Protection of the Faculty of Medicine of the University of Leipzig, datasets were not uploaded in a public repository in order not to

violate the confidentiality of the subjects' information. Datasets contain sensitive information, that might otherwise identify participants. Reasoned requests to access datasets can be directed to the Research Lab at the Department of Psychosomatic Medicine and Psychotherapy, University of Leipzig (forschung-psom@medizin.uni-leipzig.de), which serves as the permanent data access committee. Requests will be submitted to the data protection coordinator of the University of Leipzig as well as to the Ethics Committee before transfer. Based on the EU General Data Protection Regulation and the patient consent form used in the study, only de-identified data may be shared.

**Funding:** The author(s) received no specific funding for this work.

**Competing interests:** The authors have declared that no competing interests exist.

## Conclusions

This study shows COVID-19 pandemic impacts on the grieving process of relatives of patients with cancer. Consequently, screening for pre-loss grief, preparedness and their associated factors may help provide early support for relatives of people with cancer at need. However, further research is needed to help understand the stability of pre-loss grief and preparedness.

## Introduction

Cancer is one of the leading causes of death in the world [1]. In 2020, nearly 10 million people worldwide died due to cancer and 19.3 million new cases of cancer were reported [1]. For most people, the loss of a loved one to cancer is not unexpected, leaving time to adapt to the imminent death [2]. This period of forewarning has been described as grief before death—specifically anticipatory grief or more recently as pre-loss grief—and preparedness in the literature [3, 4]. Anticipatory grief was initially viewed as a resource in the sense of preparation for the loss and detachment, reducing psychological stress after the loss [5]. Because anticipatory grief has failed to fulfill the originally hypothesized function of anticipating the grieving process, Nielsen et al. [3] recommended the use of "pre-loss grief", which will be used throughout this paper. In contrast to anticipatory grief indicating a function of this form of grief, pre-loss grief only describes grief before death [3].

Literature divides preparedness into two different operationalizations: preparedness for caregiving and preparedness for death. While preparedness for caregiving is characterized as the perception of readiness regarding caregiving activities for a patient [6], preparedness for death indicates how ready relatives feel for the death of their loved one [7]. Moreover, Hebert et al. [8] described that preparedness for death consists of affective, behavioral and cognitive factors. While preparedness for caregiving has a specific scale and has been frequently examined in previous studies [6, 9], preparedness for death represents a newer concept that lacks a consistent operationalization [4].

A recent review investigated caregivers of patients with cancer and found that high levels of pre-loss grief were a risk factor for poor adaption to the death and high levels of preparedness a protective factor for poor adaptation to the death [4]. Because both constructs influence the adjustment after death, it seems important to examine predictors of pre-loss grief and preparedness to provide insight into the grieving process and therefore develop specific interventions.

Previous literature has examined a variety of predictors of pre-loss grief and preparedness. Studies show that coping style, relationship quality, social support, health status and attachment style seem to be related to levels of pre-loss grief and/or preparedness [2, 10–16]. However, many studies show ambivalence regarding the direction of the effect, such as in the case of relationship quality, as some studies found a positive and others a negative association between quality of relationship and grief or preparedness [12, 17, 18]. Moreover, most studies only assessed preparedness for death or preparedness for caregiving, not exploring both constructs at the same time. Furthermore, there seem to be important variables that may influence pre-loss grief and preparedness that have not been studied yet. One example are COVID-19 related fears. While many studies assume negative effects of COVID-19 on the grieving process [19], studies investigating the effect of COVID-19 on bereavement mostly consist of expert opinions, commentaries or recommendations for research and practice, lacking of empirical

studies [see 20]. Also, the individual prognosis of the loved ones may play a role in grieving and preparation processes [21, 22], however, its relationship with pre-loss grief or preparedness has not been studied yet.

Moreover, the aforementioned studies on factors related with pre-loss grief and preparedness mostly measured preparedness for death retrospectively with a single-item, not taking into account the different components described by Hebert et al. [see 4, 8]. Therefore, prospective studies on predictors of pre-loss grief and preparedness, that consider the different components of preparedness and distinguish between different conceptualizations of preparedness, are important.

This is, to our knowledge, the first empirical study to investigate pre-loss grief, preparedness for death and preparedness for caregiving simultaneously during the COVID-19 pandemic in relatives of people with cancer. It aims to explore various correlates of pre-loss grief and preparedness during the COVID-19 pandemic and therefore addresses current gaps in the literature. Specifically, the aims were to examine the relationship between coping strategies, quality of relationship, prognosis, COVID-19 related fears, health status, attachment style and: (1) pre-loss grief, (2) the multidimensional construct of preparedness for death and (3) preparedness for caregiving in relatives of cancer patients during the COVID-19 pandemic.

## Materials and methods

### Procedure

The study was conducted according to the Declaration of Helsinki and was approved by the Ethics Committee of the Medical Faculty of the University of Leipzig (reference number: 046/20-ek). We recruited participants between February 2020 and September 2021 via Internet, social media networks and health care providers for a cross-sectional survey (e.g., through facebook help groups and advertisements, hospices and service counseling centers for cancer information). For this purpose, we linked a study page with information about the background, objectives, and implementation of the study, which participants could read and download. Interested participants could start the survey on their own by filling out self-report measures. We integrated an electronic consent form in the first page of the online questionnaire. Due to the difficulty of accessibility of relatives of patients with cancer (as there is no official cancer registry with contact details of relatives), convenient sampling was used to ensure a sufficient sample size.

### Participants

Participants were included if they were relatives of people with cancer, 18 years or older, spoke German and provided electronic informed consent. All data entered after providing informed consent was saved automatically by the survey platform. Most participants were women (90.3%), married (56.2%), highly educated (12+years, 62.2%), had a German nationality (96.3%) and belonged to a religion (57.9%). The mean age of the participants was 41.35 years ($SD$ = 12.21). The patients with cancer were mostly parents (42.1%), partners (26.8%) or children (11.0%) of the participants. They were mostly women (53.5%) and the mean age was 54.25 years ($SD$ = 19.97). Regarding the care for the person with cancer, 41.8% stated to undertake some type of care activities.

### Measures

For this study, the main outcomes were pre-loss grief and preparedness. Pre-loss grief was assessed using the Caregiver Grief Scale [23], containing 11 items. Participants are asked to

answer on a 5-point Likert-scale (1 = strongly disagree; 5 = strongly agree). Higher scores indicate higher pre-loss grief. Internal consistency was good in this study ($\alpha$ = .82).

Preparedness measures included measures for preparedness for death and preparedness for caregiving. Questions for preparedness for death were self-generated, based on Schulz et al. [24], taking the different components suggested by Hebert et al. [8] into account. The three items can be rated on a 4-point Likert-scale (1 = not at all; 4 = very). Higher scores indicate higher preparedness for death. In the present study, Cronbach's Alpha for preparedness for death was questionable ($\alpha$ = .69). However, because different and overlapping terminology is used to describe internal consistency [see 25] and values above .7 and .6 have been described as acceptable [e.g., 26], the overall scale was used for analyses.

Preparedness for caregiving was measured using the Preparedness for Caregiving Scale [6]. The scale consists of eight items, which can be answered on a 5-point Likert-scale (0 = not at all prepared; 4 = very well prepared). Higher values represent a higher perceived preparedness for caregiving. Internal consistency for this scale was excellent ($\alpha$ = .91). Two psychologists (JK and JT) independently translated the English version of the Preparedness for caregiving scale into German. Both versions were compared for differences and merged by consensus into one German version. This version was then back translated by a native speaker. The German Version of the preparedness for caregiving scale is provided within the S1 Table. For correlates, we assessed sociodemographic variables of participants. All other included measurement tools are in Table 1, representing a complete list of correlates and outcomes.

## Data analyses

Statistical analyses were conducted using the Statistical Package for Social Sciences, version 25 (IBMⓇ SPSSⓇ). To examine correlates of pre-loss grief and preparedness, we performed multivariate linear regression analysis with pre-loss grief, preparedness for death and preparedness for caregiving as the dependent variables. Quality of relationship (depth, conflict), coping strategies (emotion-focused, problem-focused, dysfunctional), COVID-19 related fears, attachment style (avoidant, anxious), health status and prognosis were included as independent variables. A correlation matrix of all variables included in the analyses can be found in the (S6 Table).

## Results

### Sample description

In total, 646 potential participants started the online survey. Of these, 347 dropped out during the survey, and a total number of $n$ = 299 participants fully completed the online survey. The characteristics of the sample can be found in Table 2.

### Factors associated with pre-loss grief and preparedness

Bivariate correlations between pre-loss grief, preparedness and all correlate variables are in the (S6 Table). We conducted multivariate linear regression analyses. The underlying relationship analyzed with linear regression models was linear and all assumptions of the analysis performed were met.

### Factors associated with pre-loss grief

The regression analysis yielded a significant model (F(10,288) = 26.357, p < .001, see Table 3), of which five variables were significantly associated with pre-loss grief. A power analysis with

**Table 1. Measurement tools.**

| Construct | Instrument | Rating | | Reliability in this study |
|---|---|---|---|---|
| | | Item Number | Likert scale wording (scores) | Cronbach's α = |
| Pre-loss grief | Caregiver Grief Scale [23], German version (higher scores indicate higher pre-loss grief) | 11 | strongly disagree-strongly agree (1–5) | .82 |
| Preparedness | Preparedness for Death, self-generated items, based on Schulz et al. [24] (see S2 Table; higher scores indicate higher preparedness for death) | 3 | not at all-very (1–4) | .69 |
| | Preparedness for Caregiving Scale [6], translated (see S1 Table; higher scores indicate higher preparedness for caregiving) | 8 | not at all prepared-very well prepared (0–4) | .91 |
| COVID-19 related fears | Self-generated items (see S3 Table; higher scores indicate higher COVID-19 related fears) | 4 | not at all- a lot (1–5) | .75 |
| Relationship to person with cancer | Quality of Relationship Inventory [27], German version (higher scores indicate higher depth/conflict) | 18 in 2 subscales (depth and conflict in Relationship) | not at all-very (1–4) | .82-.91 for subscales |
| Coping Style | Brief-COPE [28], German version (higher scores indicate higher coping) | 28 in 3 subscales (emotion-focused, problem-focused, dysfunctional coping; see 11) | not at all-a lot (1–4) | .66-.74 for subscales |
| Attachment Style | Experiences in Close Relationships, Short Form [29], German version (higher scores indicate a higher anxious/avoidant attachment) | 6 in 2 subscales (anxious, avoidant attachment) | completely disagree-completely agree (1–7) | .68-.92 for subscales |
| Prognosis | Self-generated item (see S4 Table; higher scores indicate higher subjective change of death) | 1 | not at all-very likely (0–100%) | |
| Health Status | Self-generated item (see S5 Table) | 1 | no-yes (0–1) | |

G*Power 3.1. revealed a large sized effect for the prediction model of pre-loss grief ($f2 = .92$, $\alpha = .005$, power = .80).

Higher perceived depth of the relationship with the patient with cancer was associated with higher pre-loss grief ($\beta = .365$, $p < .001$). Moreover, more COVID-19 related fears ($\beta = .141$, $p = .002$) and a higher prognosis of death ($\beta = .241$, $p < .001$) were also associated with higher pre-loss grief. Higher emotion-focused coping strategies showed a negative relationship with pre-loss grief ($\beta = -.320$, $p = < .001$), while dysfunctional coping strategies showed a positive relationship with pre-loss grief ($\beta = .281$, $p < .001$). All other variables were not significantly related to pre-loss grief.

## Factors associated with preparedness for death

The regression analysis identified a significant model ($F(10,288) = 10.576$, $p < .001$, see Table 3) with five variables significantly associated with preparedness for death. Due to hetero-scedasticity, robust standard errors (HC3) were calculated. A power analysis with G*Power 3.1. revealed a medium sized effect ($f2 = .37$, $\alpha = .05$, power = .80).

A higher level of preparedness for death was associated with a higher prognosis of death ($\beta = .347$, $p < .001$), higher emotion-focused coping ($\beta = .242$, $p < .001$), lower dysfunctional coping ($\beta = -.229$, $p < .001$), lower COVID-19 related fears ($\beta = -.112$, $p = .037$) and health status ($\beta = .123$, $p = .025$).

## Factors associated with preparedness for caregiving

The regression analysis resulted in a significant model ($F(10,288) = 7.526$, $p < .001$, see Table 3), in which eight variables were significantly associated with preparedness for caregiving. A power analysis with G*Power 3.1. revealed a medium sized effect ($f2 = .27$, $\alpha = .005$,

**Table 2. Characteristics of the sample.**

|  | Variable | M / N | SD / % | Scale range |
|---|---|---|---|---|
| Demographic variables of participant | Age | 41.35 | 12.21 |  |
|  | Gender |  |  |  |
|  | Women | 270 | 90.3% |  |
|  | Men | 27 | 9.0% |  |
|  | Diverse | 2 | 0.7% |  |
|  | School education |  |  |  |
|  | Low 9 | 14 | 4.7% |  |
|  | Medium | 93 | 31.1% |  |
|  | High | 186 | 62.2% |  |
|  | Missings | 6 | 2.0% |  |
|  | Relationship to person with cancer |  |  |  |
|  | The person with cancer is my: |  |  |  |
|  | Child | 33 | 11.0% |  |
|  | Sibling | 21 | 7.0% |  |
|  | Parent | 126 | 42.1% |  |
|  | Partner | 80 | 26.8% |  |
|  | Friend | 11 | 3.7% |  |
|  | Other | 28 | 9.4% |  |
| Characteristics of the person with cancer | Age | 54.25 | 19.97 |  |
|  | Gender |  |  |  |
|  | Women | 160 | 53.5% |  |
|  | Men | 138 | 46.2% |  |
|  | Diverse | 1 | 0.3% |  |
| Pre-loss grief and Preparedness | Pre-loss grief | 3.49 | 0.77 | 1–5 |
|  | Preparedness for death | 5.96 | 2.19 | 3–12 |
|  | Preparedness for caregiving | 15.10 | 7.23 | 0–32 |

power = .80). Higher levels of preparedness for caregiving were associated with higher problem-focused coping ($\beta$ = .162, p = .009), higher emotion-focused coping ($\beta$ = .148, *p* = .017), and lower dysfunctional coping ($\beta$ = -.147, p = .009). On the other hand, lower levels of preparedness for death were associated with higher levels of an anxious attachment style ($\beta$ = -.169, p = .003), more conflicts in the relationship ($\beta$ = -.164, p = .004), less depth in the relationship ($\beta$ = .116, *p* = .048), and more COVID-19 related fears ($\beta$ = -.151, p = .006). Lastly, relatives who at the time of the survey had themselves a serious physical illness or disability or mental illness or disability, showed higher levels of preparedness for caregiving ($\beta$ = .157, p = .003). The other variables included were not significantly associated with preparedness for caregiving.

## Discussion

This is, to our knowledge, the first study to investigate factors associated with pre-loss grief and preparedness during the COVID-19 pandemic in relatives of people with cancer. Results showed that helpful coping strategies (emotion-focused or problem-focused coping) showed a negative relationship with pre-loss grief and dysfunctional coping a positive relationship with pre-loss grief. On the other hand, helpful coping strategies showed a positive relationship with preparedness measures and dysfunctional coping strategies a negative relationship with preparedness measures. This is also in line with previous studies [2, 10] and demonstrates the need of including knowledge about different coping strategies in interventions.

**Table 3. Multivariate regression analysis.**

| Model Term | Pre-loss Grief | | Preparedness for death | | Preparedness for Caregiving | |
|---|---|---|---|---|---|---|
| | β (r, 95%Cl) | p Value | β (r, 95%Cl) | p Value | β (r, 95%Cl) | p Value |
| Intercept | (.813 to 2.101) | < .001*** | (3.457 to 8.163) | < .001*** | (4.252 to 19.326) | .002** |
| Emotion-focused coping | -.320 (-.392, -.675 to -.356) | < .001*** | .242 (.205, .568 to 1.665) | < .001*** | .148 (.202, .402 to 4.132) | .017* |
| Problem-focused coping | -.029 (-.103, -.165 to .091) | .566 | .082 (.122, -.121 to .731) | .160 | .162 (.209, .507 to 3.500) | .009** |
| Dysfunctional coping | .281 (.335, .362 to .697) | < .001*** | -.229 (-.207, -1.877 to -.589) | < .001*** | -.147 (-.101, -4.591 to -.668) | .009** |
| Attachment anxiety | -.035 (-.027, -.067 to .029) | .441 | -.045 (-.098, -.253 to .112) | .450 | -.169 (-.240, -1.430 to -.304) | .003** |
| Attachment avoidance | .086 (.083, -.001 to .075) | .057 | .027 (-.059, -.106 to .172) | .641 | .107 (-.040, -.009 to .883) | .055 |
| COVID-19 related fears | .141 (.235, .010 to .040) | .002** | -.112 (-.129, -.109 to -.003) | .037* | -.151 (-.128, -.430 to -.074) | .006** |
| Prognosis | .241 (.265, .003 to .007) | < .001*** | .347 (.284, .014 to .027) | < .001*** | .055 (.018, -.010 to .032) | .315 |
| Quality of Relationship- Depth | .365 (.424, .354 to .598) | < .001*** | -.087 (-.231, -.725 to .074) | .110 | .116 (.122, .010 to 2.874) | .048* |
| Quality of Relationship- Conflict | -.010 (-.129, -.125 to .099) | .822 | -.019 (.015, -.436 to .302) | .721 | -.164 (-.185, -3.233 to -.611) | .004** |
| Health Status | -.046 (.016, -.231 to .067) | .282 | .123 (.117, .079 to 1.168) | .025* | .157 (.138, .890 to 4.380) | .003** |

Note. N = 299; Model 1-Pre-loss Grief: $R^2$ = .48, Adjusted $R^2$ = .46, $F_{(10,288)}$ = 26.357 (p < .001); Model 2-Preparedness for death: $R^2$ = .27, Adjusted $R^2$ = .24, $F_{(10,288)}$ = 10.576 (p < .001); Model 3-Preparedness for Caregiving: $R^2$ = .21, Adjusted $R^2$ = .18, $F_{(10,288)}$ = 7.526 (p < .001).

* p < .05

** p < .01

*** p < .001.

Regarding attachment style, only an anxious attachment style was negatively associated with preparedness for caregiving. Contrary to previous studies [12, 16], results did not show a significant relationship between attachment style and pre-loss grief and attachment avoidance and preparedness for caregiving. However, Sörensen and colleagues' [16] sample consisted of parents of college students, who did not have a loved one suffering from cancer, limiting comparability. Moreover, Pote and Wright [12] evaluated caregivers of individuals with dementia, assessing pre-loss grief with a questionnaire specifically designed for caregivers of people with dementia or Alzheimer's disease. Relatives of cancer and dementia patients show dissimilarities in the experience of pre-loss grief [30], which may account for the found differences. Furthermore, attachment style may not play a role for preparedness for death, as results did not show a significant relationship and no further studies on these relationships exist.

Additionally, more COVID-19 related fears were associated with a higher level of pre-loss grief and preparedness for caregiving, but a lower level of preparedness for death. This suggests that the COVID-19 pandemic has a significant impact on the grieving and caregiving process before a loss and represents another stress factor that should be considered in patient care. Therefore, this is the first study to confirm previous theoretical assumptions about the negative effect of the COVID-19 pandemic on the grieving process [19, 20]. As COVID-19 related fears also emerged as a significant correlate for preparedness for death, it may further complicate preparation processes for the death of a loved one.

Furthermore, relatives with a worse health status showed higher levels in both preparedness measures, which is contrary to previous results [2, 15]. This could be due to the fact that previous studies assessed health status with continuous questionnaires for a variety of health variables, e.g., general health, depression or anxiety. In contrast, health status was assessed using a dichotomous variable in this study, only measuring whether or not relatives had a physical/mental illness or disability, therefore limiting comparability to previous studies. While relatives who have a physical or mental illness/disability may have higher preparedness for death and caregiving, this relationship may dependent on the severity of their illness or disability. To further explore the relationship between health status and pre-loss grief/preparedness measures, more studies are necessary.

Moreover, results regarding quality of relationship showed that a higher perceived depth [= importance of the relationship, see 27] of the relationship with the patient with cancer was associated with higher pre-loss grief and higher preparedness for caregiving. More conflicts in the relationship were associated with less preparedness for caregiving. While previous studies have shown ambivalent results, they differed in measurement tools for relationship quality, as scales for self-evaluation and social support, as well as general acquaintance measures were used or relationship quality was seen as part of the construct without using a quantitative measurement [12, 17, 18]. Therefore, this is the first study to demonstrate the differential effects of perceived depth, or in other words, importance of the relationship [27] and conflict on pre-loss grief and preparedness.

Lastly, this study found a positive relationship between relatives' perceived poor prognosis of their loved one and pre-loss grief, as well as preparedness for death. This may be because the cognitive component of preparedness for death involves medical information about the patient, e.g., information on the prognosis [8]. Therefore, the information contained in the prognosis may be helpful for feeling prepared for the death. At the same time, relatives develop higher feelings of grief, the worse the prognosis of the patient with cancer. Therefore, pre-loss grief might be primarily relevant for relatives of patients with cancer with poor prognosis. For future studies, it might be interesting to explore the underlying mechanisms of these relationships.

The final models explain between 21 and 48% of variance, indicating a considerable influence of the related factors on the forewarning period. However, the cross-sectional design and exploratory analyses may limit the generalizability of results. Therefore, longitudinal studies should replicate findings with previously made hypotheses.

## Limitations

This study has several limitations. Most participants were women (90.3%) and highly educated (62.2%), which might affect the representativeness of the sample. Because pre-loss grief and preparedness have been found to be higher in women than in men [10, 15], the high percentage of women may have caused the sample to have higher scores in the outcomes, therefore possibly affecting the results. Also, a lower educational level has shown to be associated with higher levels of pre-loss grief and lower levels of preparedness [7, 31], which might also have affected the results, as grief may be higher and preparedness be lower in this sample than in samples with a lower educational background. Further research should seek to include a higher rate of men and participants from different educational backgrounds. Also, 53% of participants dropped out after starting the survey, which may create bias in the sample.

Moreover, preparedness for death was measured with a self-generated scale, which was based on Schulz et al. [24] and Hebert et al. [8], and showed to have a questionable reliability in this study. Future studies should develop a reliable assessment scale for preparedness for death, which considers its multi-component concept. Additionally, because this study focused on relatives of patients with cancer, effects could be different for other life limiting illnesses and should therefore be explored in further research.

Furthermore, as the COVID-19 pandemic changes consistently in terms of treatment options, mortality, and vaccinations, this might have an impact on COVID-19 related anxiety. Further research is needed to examine these relationships in more detail.

Lastly, because we measured pre-loss grief and preparedness only at one point in time, no conclusions can be made about the change or stability in these constructs over time and about the direction of causal effects. Future research should investigate if and how both pre-loss grief and preparedness change over time and if the changes can be predicted by other variables.

## Conclusions

This study demonstrates the unique correlate models of pre-loss grief, preparedness for death, and preparedness for caregiving in relatives of people with cancer. Screening for pre-loss grief and preparedness as well as for associated characteristics (COVID-19 related fears, dysfunctional coping strategies, depth of the relationship, attachment style, prognosis for death of patient with cancer, health status) may help identify individuals at risk and provide targeted interventions. To understand the stability or change of pre-loss grief and preparedness over time, further research is needed.

## Supporting information

**S1 Table. German version of the preparedness for caregiving scale.**
(DOCX)

**S2 Table. Self-generated questions for "Preparedness for death".**
(DOCX)

**S3 Table. Self-generated questions for "COVID-19 related fears".**
(DOCX)

**S4 Table. Self-generated item for "Prognosis".**
(DOCX)

**S5 Table. Self-generated questions for "Health Status".**
(DOCX)

**S6 Table. Correlation matrix of all variables included in the best subset analyses.**
(DOCX)

## Author Contributions

**Conceptualization:** Viktoria Schmidt, Julia Kaiser, Julia Treml.

**Data curation:** Viktoria Schmidt, Julia Kaiser.

**Formal analysis:** Viktoria Schmidt.

**Funding acquisition:** Anette Kersting.

**Investigation:** Julia Kaiser.

**Methodology:** Viktoria Schmidt.

**Project administration:** Anette Kersting.

**Supervision:** Anette Kersting.

**Writing – original draft:** Viktoria Schmidt.

**Writing – review & editing:** Viktoria Schmidt, Julia Kaiser, Julia Treml, Anette Kersting.

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
