## [Decision Letter · Decision Letter 0]

4 Oct 2022

PONE-D-22-17257Factors associated with pre-loss grief and preparedness in relatives of people with cancer during the COVID-19 pandemic: A cross-sectional study.PLOS ONE

Dear Dr. Schmidt,

Thank you for submitting your manuscript to PLOS ONE. After careful consideration, we feel that it has merit but does not fully meet PLOS ONE’s publication criteria as it currently stands. Therefore, we invite you to submit a revised version of the manuscript that addresses the points raised during the review process.

I recommend further revisions of the manuscript in line with the suggestion of both reviewers (especially Reviewer 2). Also make the manuscript more concise.  

We look forward to receiving your revised manuscript.

Kind regards,

Jibril Mohammed, BSc, MSc, PhD

Academic Editor

PLOS ONE

Journal Requirements:

Additional Editor Comments (if provided):

I invite you to make further corrections to your manuscript as raised by the reviewer(s).

Reviewers' comments:

Reviewer's Responses to Questions

**Comments to the Author**

1. Is the manuscript technically sound, and do the data support the conclusions?

Reviewer #1: Yes

Reviewer #2: Yes

2. Has the statistical analysis been performed appropriately and rigorously? 

Reviewer #1: Yes

Reviewer #2: Yes

3. Have the authors made all data underlying the findings in their manuscript fully available?

Reviewer #1: Yes

Reviewer #2: Yes

4. Is the manuscript presented in an intelligible fashion and written in standard English?

Reviewer #1: Yes

Reviewer #2: Yes

5. Review Comments to the Author

Reviewer #1: Please see the attached file.

Reviewer #2: Abstract.

Objectives: Please be clear on what you mean by constructs in the following phase ‘and their relationship with COVID-19 specific constructs.’ Remember the abstract is the gateway to your article.

Method: It should be ‘Data of 299 participants from a cross-sectional study was used.

Please be very clear about the analysis. What did you use Multivariate linear regression analysis for? What were the independent and the dependent variables?

From what I understand the dependent variables are Pre-loss grief, Preparedness for death, and Preparedness for caregiving; while the independent variables are perceived depth of the relationship, COVID-19 related fears, prognosis for death, dysfunctional coping strategies, emotion-focused coping strategies, prognosis for death, problem-focused coping, attachment anxiety and perceived conflict in the relationship with the cancer patient. So if I am right, in reporting your result, you should say the independent variable (name) significantly or not predicted the dependent variable (name).

Also please use patients with cancer in place of cancer patients

I think it will be good if you can include r values and the 95% CI together with the β and p values. This will make the result more meaningful

Conclusion: It should be ‘COVID-19 pandemic impacts on the grieving process of relatives of patients with cancer. Consequently, screening for pre-loss grief, preparedness and their associated factors may help provide early support for relatives of people with cancer at need. However, further research is needed to help understand the stability of pre-loss grief and preparedness.

Introduction

Although the introduction has provided many relevant information, it is way too long, and it has included many irrelevant information. I think it will be good if the authors will talk about cancer in one paragraph and how death is expected in the patients, and then connect this with pre-loss grief and preparedness in the relatives of the patients in the second paragraph. The third paragraph can talk about the problem and the aim of the study. This way, the introduction will be more succinct.

Procedure

Line 144: It should be ‘and was approved’

Page 7

Please take the following to the result section: In total, 646 potential participants started the online survey. Of these, 273 dropped out during or after completing the sociodemographic variables. Further 74 participants were excluded due to dropping out during the remaining online survey, leading to a total number of n=299 participants.

What you need under this section: are inclusion and exclusion criteria, the characteristics of the participants such as being a relative of patients with cancer, participants who can speak German, etc, etc.

Measures

I do not know why you need Table 1? What we need to see under measures are the description, characteristics and the psychometric properties of the outcome measures from previous studies, unless they are new measures invented by you.

Statistical analysis

Because three different models were tested and to correct for Type 1 error, we applied Bonferroni correction and set the significance level at p<.0167. I am not sure Bonferoni correction was appropriately applied. Bonferonni correction is used when there is multiple pairwise comparison, and in the first, you are not comparing anything with anything.

Table 3: You need to provide r values (rank order correlation) in the table. The 95% CI is for the r value.

Also please cross-check the p values in the text. Some are reading p<000

Factors associated with preparedness for caregiving

You mentioned power analysis here: A power analysis with G*Power 3.1. revealed a medium sized effect (f2 = .22, α = .0167, power = .80). What is it for?

Limitation

This is not limitation: Furthermore, the cross-sectional design and exploratory analyses may limit the generalizability of results.

Please take of the limitation to the discussion section. They are not limitations.

Conclusion: The conclusion is too long. A conclusion should be very succinct.

6. PLOS authors have the option to publish the peer review history of their article (what does this mean?). If published, this will include your full peer review and any attached files.

Reviewer #1: **Yes: **Naziru Bashir Mukhtar

Reviewer #2: **Yes: **Auwal Abdullahi

---

## [Author Response · Author response to Decision Letter 0]

7 Nov 2022

PONE-D-22-17257

Factors associated with pre-loss grief and preparedness in relatives of people with cancer during the COVID-19 pandemic: A cross-sectional study.

Dear Prof. Dr. Prof. Chenette,

We would like to thank the editor and the reviewers for their effort in reviewing our manuscript “Factors associated with pre-loss grief and preparedness in relatives of people with cancer during the COVID-19 pandemic: A cross-sectional study”. We appreciated the reviewers' constructive comments, which helped us to improve the quality of the manuscript with regard to clarity and conciseness. We have considered the reviewers’ comments and responded in detail to each of them. Please find these answers below. Furthermore, we revised the manuscript to be in line with all modifications. All modifications, which address the suggestions, are highlighted in the paper. 

Reviewer comments

Reviewer 1

We would like to thank the reviewer for her/his detailed comments and suggestions, which helped us to improve our manuscript.

Line 50. Include the reference for first sentence.

• Thank you for bringing this to our attention, we included a reference for the first sentence (page 4, line 75).

Line 61-68. The paragraph contains in-text citations with page numbers, I am not sure if that conforms with PLOS ONE guidelines.

• Thank you for bringing this to our attention. We rephrased and removed in-text citations, as well as page numbers (page 4, line 87-96).

Line 71. ….. were a risk factor (for what?)

• We included further information to the sentence: “A recent review investigated caregivers of cancer patients and found that high levels of pre-loss grief were a risk factor for poor adaption to the death and high levels of preparedness a protective factor for poor adaptation to the death (4).” (page 4, line 98).

Page 125-132 H1-H3 The hypotheses should be in future tenses. …will be associated… not ….is associated 

Page 133-134 Furthermore, we assume that quality of relationship to the patient and prognosis of the patient are (use future tense, will be) associated

……

• Thank you for mentioning this. We agree and rephrased the hypotheses in future tense. However, to make the introduction more succinct and in accordance with the comments of the first reviewer, we removed the hypotheses from the study (page 4-5).

Page 146-7 ..social networks? You mean social media networks?

• Thank you for bringing this to our attention. You are right. We therefore rephrased it to “social media networks” (page 6, line 139).

Page 152-153 Why the use of convenient sampling?

• Thank you for pointing this out. We added this information to the manuscript: “Due to the difficulty of accessibility of relatives of cancer patients (as there is no official cancer registry with contact details of relatives), convenient sampling was used to ensure a sufficient sample size.” (page 6, line 145-147).

Page 158-159 Were the participants able to submit the survey after completing only demographic information? How dropouts were obtained is not clearly stated.

• Thank you for mentioning this. All data entered was automatically saved by the survey platform after electronic informed consent was provided, regardless if participants decided to drop out or finish the questionnaire. So when participants entered only demographic information and closed the survey, the previously entered data was automatically saved by the survey platform. We also added this information to the manuscript: “All data entered after providing informed consent was saved automatically by the survey platform.” (page 6-7, line 154-155).

Page 159-160 How many times were the survey(s) completed? You reported additional dropouts during the remaining online surveys

• Thank you for pointing this out. The survey was only completed during one time. As the information was not clearly described we changed the sentence: “In total, 646 potential participants started the online survey. Of these, 347 dropped out during the survey, and a total number of n=299 participants fully completed the online survey.” (page 9, line 204-205).

Page 162 Was written informed consent really provided? I thought they indicated their consent in the online survey link. Maybe you rewrite?

• Thank you for bringing this to our attention. We rephrased “written informed consent” to “electronic informed consent” (page 6, line 154)

164 Measures sub heading. For all the scales, did you use the German versions or you translated them? Give details. Also, for Table 1, is it possible to indicate which scores indicate better outcomes for the rest of the scales? You singled out a few in the procedure and mention higher scores indicates………

• Thank you for mentioning this. We agree and added in Table 1 for every scale, if it was a German version, self-generated or translated. We also added for each scale, what higher scores indicate. (page 8, Table 1)

• Following scale was translated: Preparedness for Caregiving. We added this information in the manuscript: “The German Version of the preparedness for caregiving scale is provided within the supplementary material. Two psychologists (JK and JT) independently translated the English version of the Preparedness for caregiving scale into German. Both versions were compared for differences and merged by consensus into one German version. This version was then back translated by a native speaker. The German Version of the preparedness for caregiving scale is provided within the supplementary material (S1 Table)” (page 7-8, line 179-184).

Page 233…… while the opposite was true…… the statement can confuse the reader in interpreting the result. Use straightforward language 

• Thank you for bringing this to our attention. We changed the sentence to: “Higher emotion-focused coping strategies showed a negative relationship with pre-loss grief (β=-.319, p=<.001), while dysfunctional coping strategies showed a positive relationship with pre-loss grief (β=.279, p<.001).” (page 11, line 240).

Page 235 Table 3. Is it possible to indicate p-values that are significant? It makes reading through easy 

• Thank you for mentioning this. We agree and indicated in Table 3, which p-values were significant (Table 3, page 12-13, line 245).

Page 261 under discussion. The sentence that starts with ‘In accordance with our first hypothesis…..’ needs corrections

• We agree and rephrased the sentence to: “Results showed that helpful coping strategies (emotion-focused or problem-focused coping) showed a negative relationship with pre-loss grief and dysfunctional coping a positive relationship with pre-loss grief. On the other hand, helpful coping strategies showed a positive relationship with preparedness measures and dysfunctional coping strategies a negative relationship with preparedness measures.” (page 15, line 276-280).

Page 267 Regarding our second hypothesis (not hypotheses)

• Thank you for mentioning this. This sentence was removed in accordance with the comments of the second reviewer.

Page 309 under limitations. It will be good to discuss how that women dominance in the sample might have affected the result. Are there studies indicating which gender is associated with more grief and other outcomes of concern in this study? The same goes to literacy level. Majority of the participants are highly educated. Any literature on literacy level and the outcomes of this study?

• Thank you for mentioning this. We included in the discussion how gender and educational background may have affected the sample: “Because pre-loss grief and preparedness have been found to be higher in women than in men (26,37), the high percentage of women may have caused the sample to have higher scores in the outcomes, therefore possibly affecting the results. Also, a lower educational level has shown to be associated with higher levels of pre-loss grief and lower levels of preparedness (7,38), which might also have affected the results, as grief may be higher and preparedness be lower in this sample than in samples with a lower educational background.” (page 17, line 341-347).

Did you collect health information of the participants themselves? Don’t you think their health status may also affect their responses?

• Thank you for bringing this to our attention. You are right that relatives’ health status may influence the outcome variables. As health status was assessed in our study, we included it as a further predictor. Therefore, we recalculated our analysis and rewrote our manuscript regarding health status: 

o “Studies show that coping style, relationship quality, social support, health status and attachment style seem to be related to levels of pre-loss grief and/or preparedness.” (page 5, line 104-105)

o Table 3 (page 12)

o “A higher level of preparedness for death was associated with a higher prognosis of death (β=.358, p<.001), higher emotion-focused coping (β=.241, p<.001), lower dysfunctional coping (β=-.222, p<.001), lower COVID-19 related fears (β=-.112, p=.037) and health status (β=.123, p=.025).” (page 14, line 251-254) 

o “Lastly, relatives who at the time of the survey had themselves a serious physical illness or disability or mental illness or disability, showed higher levels of preparedness for caregiving (β=.157, p=.003).” (page 14, line 265-268).

o “Furthermore, relatives with a worse health status showed higher levels in both preparedness measures, which is contrary to previous studies (2,15). This could be due to the fact that previous studies assessed health status with continuous questionnaires for a variety of health variables, e.g., general health, depression or anxiety. In contrast, health status was assessed using a dichotomous variable in this study, only measuring whether or not relatives had a physical/mental illness or disability, therefore limiting comparability to previous studies. While relatives who have a physical or mental illness/disability may have higher preparedness for death and caregiving, this relationship may dependent on the severity of their illness or disability. To further explore the relationship between health status and pre-loss grief/preparedness measures, more studies are necessary.” (page 16, line 305-314).

o Supporting Information: S5_Table

Another potential limitation, did you ask how much information the participants know about the cancer patients? The amount of knowledge they have will definitely play a role in their grief and preparedness 

• Thank you for mentioning this. We agree that the amount of information plays an important role for relatives of patients with cancer. However, as the amount of information is included in our measurement for preparedness for death (“If the sick person were to die soon, would you already have all the information you need?”, S2_Table), therefore we did not include this as an extra predictor in our analysis. 

REVIEWER 2

We would like to thank the reviewer for her/his detailed comments and suggestions, which helped us to improve our manuscript.

Abstract.

Objectives: Please be clear on what you mean by constructs in the following phase ‘and their relationship with COVID-19 specific constructs.’ Remember the abstract is the gateway to your article.

• Thank you for mentioning this. We agree and rephrased “COVID-19 specific constructs” to “COVID-19 related fears” (Page 2, line 28)

Method: It should be ‘Data of 299 participants from a cross-sectional study was used.

Please be very clear about the analysis. What did you use Multivariate linear regression analysis for? What were the independent and the dependent variables?

• Thank you for bringing this to our attention. We agree and rephrased the Method-section of the abstract to be more clear: “Data of 299 participants from a cross-sectional study was used. Participants were included if they were relatives of people with cancer, spoke German and were at least 18 years. Multivariate linear regression analyses were conducted to measure the relationship between predictors (dysfunctional coping, emotion-focused coping, problem-focused coping, attachment anxiety, attachment avoidance, COVID-19 related fears, prognosis, perceived depth of the relationship, perceived conflict in the relationship, health status) and pre-loss grief, preparedness for caregiving and preparedness for death as the dependent variables.” (page 2, line 29-36).

From what I understand the dependent variables are Pre-loss grief, Preparedness for death, and Preparedness for caregiving; while the independent variables are perceived depth of the relationship, COVID-19 related fears, prognosis for death, dysfunctional coping strategies, emotion-focused coping strategies, prognosis for death, problem-focused coping, attachment anxiety and perceived conflict in the relationship with the cancer patient. So if I am right, in reporting your result, you should say the independent variable (name) significantly or not predicted the dependent variable (name).

• Thank you for bringing this to our attention. We agree and rephrased the Result-section of the abstract into saying which independent variable significantly predicted the dependent variable: “Perceived depth (β=.365, p<.001), COVID-19 related fears (=.141, p=.002), prognosis for death (β=.241, p<.001), dysfunctional coping strategies (β=.281, p<.001) and emotion-focused coping strategies (β=-.320, p<.001) significantly predicted pre-loss grief. Prognosis for death (β=.347, p<.001), dysfunctional coping strategies (β=-.229, p<.001), emotion-focused coping strategies (β=.242, p<.001), COVID-19 related fears (β=-.112, p=.037) and health status (β=.123, p=.025) significantly predicted preparedness for death. Dysfunctional coping (β=-.147, p=.009), problem-focused coping (β=.162, p=.009), emotion-focused coping (β=.148, p=.017), COVID-19 related fears (β=-.151, p=.006), attachment anxiety (β=-.169, p=.003), perceived conflict in the relationship with the patient with cancer (β=-.164, p=.004), perceived depth in the relationship (β=.116, p=.048) and health status (β=.157, p=.003) significantly predicted preparedness for caregiving.” (page 2, line 37-47).

• Unfortunately, due to word limit of the abstract (a maximum of 300 words), it was not possible to include the remaining independent variables, that did not significantly predict the outcome variables.

Also please use patients with cancer in place of cancer patients

• Thank you for mentioning this. We replaced “cancer patients” with “patients with cancer” throughout the manuscript.

I think it will be good if you can include r values and the 95% CI together with the β and p values. This will make the result more meaningful

• Thank you for mentioning this. We agree that including the r values and the 95%Cl would make the results more meaningful. However, due to the word limit of the abstract (a maximum of 300 words), it was not possible to include this information in the abstract. Nevertheless, Table 3 (page 12) in the manuscript includes the β, p, r values and 95% Cl.

Conclusion: It should be ‘COVID-19 pandemic impacts on the grieving process of relatives of patients with cancer. Consequently, screening for pre-loss grief, preparedness and their associated factors may help provide early support for relatives of people with cancer at need. However, further research is needed to help understand the stability of pre-loss grief and preparedness.

• Thank you for pointing this out. We rephrased the conclusion to match the aforementioned conclusion: “This study shows COVID-19 pandemic impacts on the grieving process of relatives of patients with cancer. Consequently, screening for pre-loss grief, preparedness and their associated factors may help provide early support for relatives of people with cancer at need. However, further research is needed to help understand the stability of pre-loss grief and preparedness.” (page 3, line 57-61).

Introduction

Although the introduction has provided many relevant information, it is way too long, and it has included many irrelevant information. I think it will be good if the authors will talk about cancer in one paragraph and how death is expected in the patients, and then connect this with pre-loss grief and preparedness in the relatives of the patients in the second paragraph. The third paragraph can talk about the problem and the aim of the study. This way, the introduction will be more succinct.

• Thank you for bringing this to our attention. We agree and rephrased and abbreviated the introduction to match these three paragraphs and make it more succinct (Introduction, page 4-6)

Procedure

Line 144: It should be ‘and was approved’

• Thank you for bringing this to our attention. We changed it to “was approved” (page 6, line 136).

Page 7

Please take the following to the result section: In total, 646 potential participants started the online survey. Of these, 273 dropped out during or after completing the sociodemographic variables. Further 74 participants were excluded due to dropping out during the remaining online survey, leading to a total number of n=299 participants.

• Thank you for mentioning this. We moved the above mentioned paragraph from the Method to the Result section (page 9, line 205-206). 

What you need under this section: are inclusion and exclusion criteria, the characteristics of the participants such as being a relative of patients with cancer, participants who can speak German, etc, etc.

• Thank you for pointing this out. We moved the characteristics of the participants to the Method-section (page 6-7, line 153-161).

Measures

I do not know why you need Table 1? What we need to see under measures are the description, characteristics and the psychometric properties of the outcome measures from previous studies, unless they are new measures invented by you.

• It is true that that Table 1 is not necessarily needed for understanding the manuscript, however, it provides a good overview for readers. Following the advice from the second reviewer, we kept Table 1 in the manuscript and added more information (page 8, Table 1).

Statistical analysis

Because three different models were tested and to correct for Type 1 error, we applied Bonferroni correction and set the significance level at p<.0167. I am not sure Bonferoni correction was appropriately applied. Bonferonni correction is used when there is multiple pairwise comparison, and in the first, you are not comparing anything with anything.

• Thank you for bringing this to our attention. We removed the Bonferroni correction from our analysis and manuscript. As a result, more variables became significant in predicted the outcome variables. This can be seen in Table 3 (page 12).

Table 3: You need to provide r values (rank order correlation) in the table. The 95% CI is for the r value.

• Thank you for mentioning this. We included the rank order correlations in Table 3 (page 12). 

Also please cross-check the p values in the text. Some are reading p<000

• Thank you for pointing this out, we cross-checked the p-values in the text and removed errors from the manuscript.

Factors associated with preparedness for caregiving

You mentioned power analysis here: A power analysis with G*Power 3.1. revealed a medium sized effect (f2 = .22, α = .0167, power = .80). What is it for?

• Thank you for mentioning this. We added this information to the manuscript: “A power analysis with G*Power 3.1. revealed a large sized effect for the prediction model of pre-loss grief (f2 = .92, α = .005, power = .80).” (page 11, line 233-234).

Limitation

This is not limitation: Furthermore, the cross-sectional design and exploratory analyses may limit the generalizability of results.

Please take of the limitation to the discussion section. They are not limitations. 

• Thank you for bringing this to our attention. We removed this section from the limitation to the discussion section (page 17, line 335-337).

Conclusion: The conclusion is too long. A conclusion should be very succinct.

• Thank you for mentioning this. We agree and abbreviated the conclusions to make it more succinct (page 18, line 369-383).

---

## [Editor Report · Decision Letter 1]

14 Nov 2022

Factors associated with pre-loss grief and preparedness in relatives of people with cancer during the COVID-19 pandemic: A cross-sectional study.

PONE-D-22-17257R1

Dear Dr. Schmidt,

We’re pleased to inform you that your manuscript has been judged scientifically suitable for publication and will be formally accepted for publication once it meets all outstanding technical requirements.

Kind regards,

Jibril Mohammed, BSc, MSc, PhD

Academic Editor

PLOS ONE
---

## [Editor Report · Acceptance letter]

17 Nov 2022

PONE-D-22-17257R1 

Factors associated with pre-loss grief and preparedness in relatives of people with cancer during the COVID-19 pandemic: A cross-sectional study. 

Dear Dr. Schmidt:

I'm pleased to inform you that your manuscript has been deemed suitable for publication in PLOS ONE. Congratulations! Your manuscript is now with our production department. 

Kind regards, 

on behalf of

Dr. Jibril Mohammed 

Academic Editor

PLOS ONE